# Control of randomly scattered surface plasmon polaritons for multiple-input and multiple-output plasmonic switching devices

Wonjun Choi[1,2,*], Yonghyeon Jo[1,2,*], Joonmo Ahn[1,2,3], Eunsung Seo[1,2], Q-Han Park[2], Young Min Jhon[3] & Wonshik Choi[1,2]

Merging multiple microprocessors with high-speed optical networks has been considered a promising strategy for the improvement of overall computation power. However, the loss of the optical communication bandwidth is inevitable when interfacing between optical and electronic components. Here we present an on-chip plasmonic switching device consisting of a two-dimensional (2D) disordered array of nanoholes on a thin metal film that can provide multiple-input and multiple-output channels for transferring information from a photonic to an electronic platform. In this device, the surface plasmon polaritons (SPPs) generated at individual nanoholes become uncorrelated on their way to the detection channel due to random multiple scattering. We exploit this decorrelation effect to use individual nanoholes as independent antennas, and demonstrated that more than 40 far-field incident channels can be delivered simultaneously to the SPP channels, an order of magnitude improvement over conventional 2D patterned devices.

[1] Center for Molecular Spectroscopy and Dynamics, Institute for Basic Science, Seoul 02841, Korea. [2] Department of Physics, Korea University, Seoul 02855, Korea. [3] Sensor System Research Center, Korea Institute of Science and Technology, Seoul 02792, Korea. * These authors contributed equally to this work. Correspondence and requests for materials should be addressed to Wk.C. (email: wonshik@korea.ac.kr).

The clock speed of microprocessors has steadily grown over the past few decades, but it has now been saturated at around a few gigahertz. Merging multiple microprocessors has been one of the main strategies employed to increase overall computation power, but the electrical networks connecting the processors have been unable to keep up with the data streams that the individual processors produce. Optical interconnects have been considered potential solutions because optical communication is hundreds of times faster than that of electrical networks. For coupling optics to electronics, surface plasmon polaritons (SPPs) are ideal carriers because they propagate as electromagnetic waves along the dielectric/metal interface and yet induce collective charge oscillations in a metal[1–3]. For this reason, the use of SPPs for on-chip optical interconnects has gained interest in recent years[4–12], although they have long been applied in biosensing[13–16], lithography[17–19], subwavelength imaging[20,21] and nanomanipulation[22,23] due to their subwavelength-scale spatial confinement and strong local field enhancement[24].

In designing the SPP-based optoelectronic devices, it is important to implement multi-channel optical inputs to multi-channel plasmonic outputs for the maximized use of optical communication bandwidth. Otherwise, the devices will serve as bottlenecks in the data transfer. However, it is not straightforward to effectively convert the input channels of far-field waves propagating in three-dimensional space to the output channels of the two-dimensional (2D) surface waves. In fact, this dimensional reduction leads to the loss of most input channels upon conversion to output channels. In recent years, various multiplexing methods have been proposed to increase the deliverable channel number. For instance, structures have been designed in such a way that either the directionality[25] or the beam shape of the output SPPs is controlled by the polarization[23], wavelength[26] and intensity patterns of incident waves[22]. While these studies demonstrated great potential, the number of controllable transmission channels remains small mainly due to the use of periodically ordered structures. In these cases, SPPs generated at each unit cell are replicated at the other unit cells such that they are not independent from one another. While this strategy has been necessary to increase the coupling of incident far-field waves to the SPPs by means of the constructive interference of SPPs generated among unit cells, it is ineffective in the context of transferring input channels to the SPP channels.

In this article, we propose the use of a 2D disordered array of nanoholes patterned on a thin metal film[27] to increase the number of transmission channels from far-field optical inputs to plasmonic outputs. Here, the disorder enables the nanoholes distributed across 2D area to act as independent antennas by means of the random multiple scattering of SPPs generated at individual nanoholes. This is clearly distinct from the case of periodically ordered 2D patterns. We prove that the effective number of deliverable channels can be more than 40, an order of magnitude increase in comparison with existing multiplexing methods. With the increased channel number, we demonstrate simultaneous focusing of SPPs at multiple spots, which is equivalent to the implementation of multiple-input and multiple-output (MIMO) networks. In addition, we show the delivery of a 2D far-field image to the SPPs.

## Results

### Channel loss from optical inputs to plasmonic outputs.
In converting the input channels of far-field waves propagating in three-dimensional space to the output channels of the surface waves, there occurs inevitable dimensional reduction that leads to the loss of most input channels. To make it clear, let us take a periodic structure on a thin metal film for an example

(Fig. 1a). The number of input channels is given by the number of diffraction-limit spots within the area of illumination, which is given by $N_{\max}^{2D} = (2L/\lambda)^2$ with $L$ the side length of the pattern, and $\lambda$ the wavelength of incident wave in free space. On the other hand, the number of output channels for the surface wave is given by the length of the output line divided by half the wavelength, that is, $2L/\lambda_{SPP}$ with $\lambda_{SPP}$ the wavelength of SPPs. Therefore, the output channels are significantly smaller in number than the input channels. In the case of a periodic structure, the channel conversion efficiency is even worse. There are only one or two output channels because SPPs generated at each unit cell, instead of acting independently, are replicated at the other unit cell.

A simple way to avoid the problem associated with the periodically patterned devices is to use a one-dimensional (1D) array of nanoholes in the metal film (Fig. 1b). Individual nanoholes are scatterers that convert the incoming far-field wave to the SPPs. Therefore, they can act as antennas sending far-field information to the detection channel. One can either choose the phase map of incident wave to individual holes (Fig. 1b)[28,29] or design the arrangement of nanoholes[30] to focus SPPs at an arbitrary spot where an electronic circuit is to be connected. In this case, however, the maximum number of transmission channels is given by $N^{1D} = 2L/\lambda$, which is the same in dimension as the SPP channels. Therefore, the dimensional reduction problem remains unsolved.

### Deliverable channel number via disordered nanohole array.
To increase the number of transmission channels from far-field optical inputs to plasmonic outputs, we propose the use of a 2D disordered array of nanoholes patterned on a thin metal film (Fig. 1c,d) (see Supplementary Note 3 for the possible layout of the optoelectronic networks). The nanoholes distributed across 2D area can act as independent antennas due to the random multiple scattering of SPPs generated at individual nanoholes. This is clearly distinct from the case of periodically ordered 2D patterns. However, the multiple scattering process is problematic because it makes the SPP fields unpredictable at the detection channels (Fig. 1c). However, there is a way to deterministically control these randomly scattered SPPs. We can measure a transfer matrix[31–33] connecting far-field input to SPP output. From this matrix, the wavefront of an incident wave that would maximize the SPPs at arbitrary target points can be identified. By shaping this wavefront using a spatial light modulator (SLM), it is possible to focus the SPPs at target spots (Fig. 1d). In our study, we defined $N_{eff}^{2D}$ as the signal-to-noise ratio of intensity at the target spot with respect to the average intensity in the background because this signal-to-noise ratio is determined by the number of independent channels transferred to the detection.

We performed theoretical analysis for the effective number of transmission channels $N_{eff}^{2D}$ for the 2D disordered array of nanoholes. The SPPs generated at the holes illuminated by a far-field wave experience multiple scattering events on their way to the sampling line (Fig. 1c). The scattering occurs in two different pathways—in-plane scattering which alters the propagation direction of the SPPs and out-of-plane scattering to the far-field waves. In-plane scattering gives rise to the decorrelation of SPPs generated at particular nanoholes with those at the other nanoholes and makes the contribution of individual nanoholes to the detection channels to be independent. Therefore, it plays an important role in increasing the number of effective transmission channels. Out-of-plane scattering works the same way as absorption loss in the sense that the photons disappear from the medium, thereby attenuating the intensity of

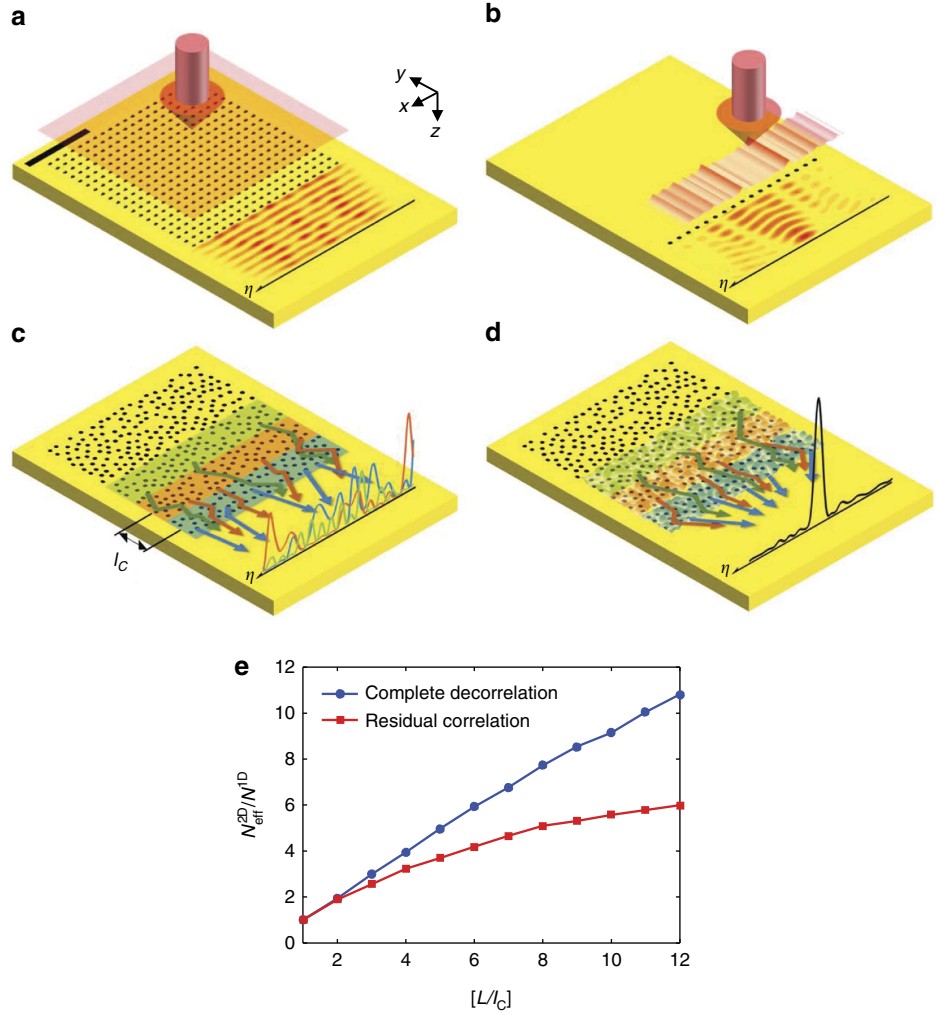

**Figure 1 | Numerical and theoretical analyses describing the performance of a 2D disordered array of nanoholes in channel transfer.** (**a**) A 2D array of periodic nanoholes patterned on a metal film. Black dots indicate the positions of the holes. SPPs generated by a normally incident plane wave propagates along $y$-direction. Scale bar, 2 μm. (**b**) A 1D array of nanoholes patterned on a metal film. The incident wave whose wavefront is properly shaped focuses the SPPs generated at the nanoholes at a target spot on a sampling line ($\eta$). (**c**) A 2D array of disordered nanoholes patterned on a metal film. Ordinary planar incident waves generate speckled SPPs. The blue, red and green curves at the sampling line are the SPP fields originating from the representative far-field illumination of the blue, red and green rectangular areas, respectively. The wavelength of the light source was 620 nm. The SPPs were uncorrelated if the centre-to-centre distance between two neighbouring illuminations was larger than the characteristic length $l_c$ described in the main text. (**d**) The same pattern of nanoholes as (**c**), but the correct choice of wavefront for the illuminations at the blue, red and green rectangular areas can cause the SPPs to constructively interfere at the target point (black curve). All the results displayed here were derived from the numerical simulations using the finite-difference time-domain (FDTD) method (see Supplementary Note 1 for details). (**e**) Expected enhancement factor for channel number $\alpha = N_{\text{eff}}^{2D}/N^{1D}$ with respect to the number of effective slabs $m = [L/l_c]$ (blue dots). The red dots were the enhancement factor predicted after accounting for the residual correlations shown in Fig. 4d (see Supplementary Note 2).

the SPPs. In fact, we could determine the scattering ($l_s$) and transport ($l_t$) mean free paths of the SPPs propagating through the disordered metal film depending on the size and fill factor of the nanoholes (see Supplementary Note 4), and use these parameters to quantitatively describe the effect of multiple scattering events on the effective number of transmission channels.

As shown in Fig. 1c, the SPPs generated at one slab of illumination gradually become uncorrelated with those at the other slabs as the separation between the slabs increases. Complete decorrelation occurs at the characteristic length $l_c$, which is mainly determined by the diffraction-limit width of the far-field illumination at a high fill factor of the nanoholes. The $l_c$ was measured to be 1.0 μm in the experiment and 0.8 μm in the numerical analysis (Supplementary Note 4), close to

the width of illumination set by the numerical aperture of 0.6 used in the experiment. Therefore, slabs of width $l_c$ can be treated as independent sources, and the 2D disordered array of nanoholes in the area $L \times L$ can be considered the combination of individual slabs of width $l_c$. The blue, green and red rectangular areas in Fig. 1c represent these slabs. Effectively, there are $m = [L/l_c]$ slabs, where [] stands for the *floor* function which independently contributes to the SPPs at the output channels. Since there are $N^{1D} = 2L/\lambda_{\text{SPP}}$ independent antennas per slab along the $x$-direction, the total number of antennas for the entire pattern is given by $m \times N^{1D}$.

The out-of-plane scattering and the metallic loss of SPPs attenuate SPP intensity depending on the distance between the holes to the sampling line, thereby reducing the effective channel number. The decay of intensity takes the form

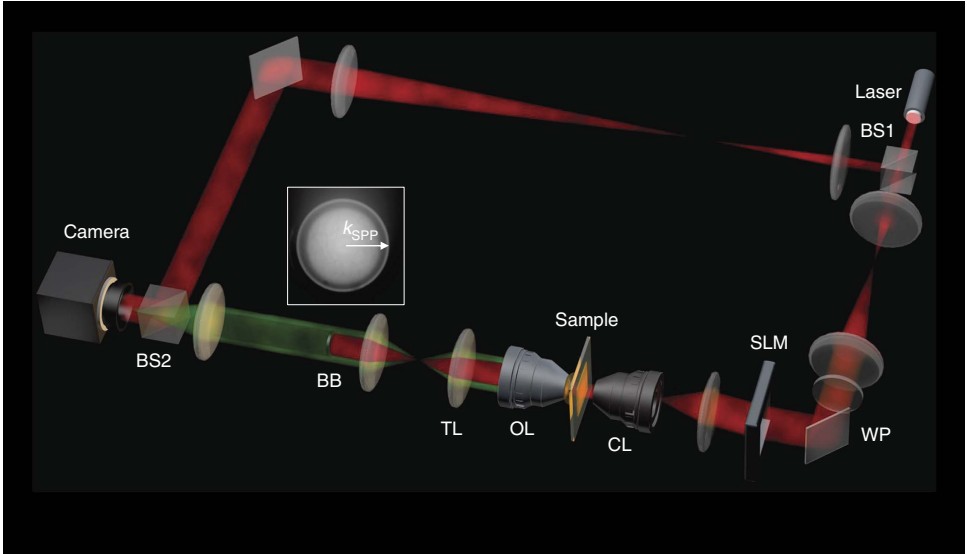

**Figure 2 | Experimental setup for the measurement of a transfer matrix and the control of SPPs.** An interferometric leakage radiation microscope equipped with a wavefront shaping device. An output beam from a helium–neon (He-Ne) laser was divided into sample and reference waves. SLM, reflection-mode spatial light modulator, but shown as transmission mode for simplicity. WP, quarter-wave plate; CL, condenser lens; OL, objective lens; TL, tube lens; BB, a circular beam block plate removing the far-field components of transmitted waves; BS1 and 2, beam splitters. The image shown above BB is the intensity image taken in front of BB. The bright sharp ring, whose radius corresponds to $k_{SPP}$, is the intensity of SPPs at the Fourier plane.

$T(y) = \frac{l_t}{y+c}\exp(-y/l_a)$ (ref. 34), where $l_a$ and $l_t$ are absorption length due to metallic loss and transport mean free path, respectively, and $c$ is extrapolation length (see the Supplementary Note 4 for experimental and numerical measurements of $T(y)$). In general, $l_a$ plays a major role at small $y$, and $l_t$ becomes more important at large $y$ (see the Supplementary Note 4). We defined the effective channel number $N_{eff}^{2D}$ by the ratio of signal intensity at the target point to the average intensity in the background (signal-to-noise ratio) for SPP focusing:

$$N_{eff}^{2D} = \frac{\left|\sum_{j=0}^{m-1}\sqrt{T(jl_c)}\right|^2}{\sum_{j=0}^{m-1}T(jl_c)}N^{1D} \equiv \alpha N^{1D}. \tag{1}$$

Here the numerator accounts for the constructive addition of SPP fields at the target point, and the denominator for the incoherent addition of SPPs at points other than the target point. The $\alpha = N_{eff}^{2D}/N^{1D}$, the enhancement factor of channel number due to the dimensional increase, is plotted as the blue dots in Fig. 1e as a function of $m = [L/l_c]$. Under the experimental conditions introduced below, the theoretical model predicts $\alpha_{th}$ to be around 9 for $L = 10\,\mu m$ and $l_c = 1\,\mu m$.

**Experimental setup**. We constructed an experimental setup to measure the transfer matrix from far-field input to SPP output (Fig. 2). The setup is based on a leakage radiation microscope[35–37], to which we added a reference wave for the recording of the phase and amplitude of the SPPs generated at the air/metal interface. In addition, we installed an SLM in the sample beam path to control the angle of the incident waves to the sample for the measurement of the transfer matrix. The device was also used to shape the wavefront of the incident waves to focus SPPs at target spots. The pixel size of the SLM was $20 \times 20\,\mu m^2$, and the magnification from SLM to the sample plane was 1/444. The number of SLM pixels used for illumination was $230 \times 230$ to match the $10 \times 10\,\mu m^2$ area where nanoholes were patterned. A quarter-wave plate was installed at the upstream of the SLM to set the polarization of the illumination

to be circular to ensure that the scattering angles of SPPs at the holes were isotropic. As a sample, we coated 100 nm-thick Au film on a glass substrate and fabricated a 2D disordered array of nanoholes using a focused ion beam (Fig. 3a). Individual holes measuring 100 nm in diameter filled an $L \times L = 10 \times 10\,\mu m^2$ area with a fill factor ranging from 3 to 15%. Since the channel conversion efficiency was the best at 12% fill factor in our experiment (see Supplementary Note 5), all the data shown in the main text used the 12% samples. The coated layer of the sample faced the condenser lens, and no immersion medium was inserted between the sample and the lens. This set the magnitude of the wavevector of the SPPs generated on the air/metal interface at $k_{SPP} = n_{SPP} \times k_0$, where $k_0 = 2\pi/\lambda$ with $\lambda$ the wavelength of the light source in a vacuum (helium–neon laser, $\lambda = 633\,nm$) and $n_{SPP} = 1.051$.

The SPPs generated at the air/metal interface were leaked toward the metal/glass interface and propagated down to the bottom of the glass substrate ($n_{glass} = 1.515$). These leaked SPPs could be picked up by an oil-immersion objective lens because the refractive index of oil ($n_{oil} = 1.515$) is larger than $n_{SPP}$. Due to the boundary conditions, the transverse wavevectors, the component of wavevectors projected to the $x$–$y$ plane of the leaked SPPs in the glass is equal to the $k_{SPP}$. Therefore, the azimuthal angle $\varphi$ of the SPP waves at the glass is given by $\varphi = \sin^{-1}(n_{SPP}/n_{glass})$. As a consequence, the SPPs appeared as a circular ring at the conjugate plane of the back focal plane of the objective lens in which the map of the Fourier transform of the transmitted electric field is displayed (inset in Fig. 2). In addition to the SPPs, far-field waves scattered by the nanostructures at the air/metal and metal/glass interfaces were also present. The transverse wavevectors of these waves can range from 0 to $n_{glass}k_0$. Therefore, the far-field waves covered the circular area with a radius corresponding to $n_{glass}k_0$, which is larger than $k_{SPP}$. However, the high spatial frequency components generated at the metal/glass interface were weak as the incident waves were significantly attenuated there. As such, far-field waves were mostly confined to a circle of radius $k_0$, mostly due to the leakage of scattered waves generated at the holes on the air/metal

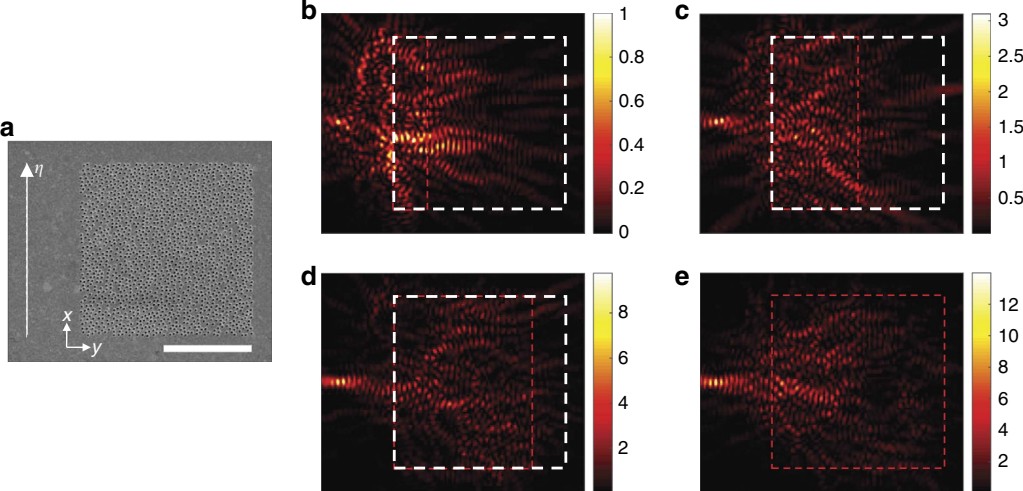

**Figure 3 | Experimental demonstration of SPP focusing. (a)** Photograph of a 2D disordered array of nanoholes patterned on Au film taken by the scanning of a focused ion beam. The positions of the individual holes are described by the coordinates $x$ and $y$, and $\eta$ indicates a coordinate along the sampling line. **(b–e)** Intensity maps for SPPs imaged at the camera when the phase maps of far-field illumination were set to maximize the intensity of the SPPs at a specific target spot on the sampling line (see main text for the identification of the appropriate phase maps). The width of illumination was $W = 2, 5, 8$ and $10\,\mu m$ for **(b–e)** respectively. The white rectangular box shows the boundary of the array of nanoholes. The red rectangular box indicates the area where the far-field waves illuminated. The colour bars indicate the intensity in arbitrary units on the same scale for **(b–e)**. Scale bar, $5\,\mu m$.

interface. We placed a circular beam block plate at the conjugate plane of the back aperture of the objective lens to block far-field waves whose transverse wavevector is smaller than $k_{SPP}$. According to our analysis, the total intensity of the residual far-field waves was $\sim 10\%$ of the total SPP intensity (Supplementary Note 5). The surface roughness of the metal layer also generated unwanted SPPs and far-field waves, but their contribution was measured to be $\sim 1\%$ of the total detected wave. An additional lens was positioned at the downstream to relay the SPP map for the sample to the camera (RedLake M3). The view field was $26 \times 26\,\mu m^2$ at the sample plane. A reference wave was linearly polarized along the $y$-direction and introduced to the camera via a beam splitter to form interference fringes with a sample wave, from which we obtained the phase and amplitude maps of the generated SPPs[38].

**Experimental demonstration of the increased channel number.**
We experimentally constructed the transfer matrix by measuring the amplitude and phase maps of SPPs for the illumination of far-field plane waves over a wide angular range. Various angles of illumination (a total of 400) were chosen in such a way that the transverse wavevectors of the plane waves formed a complete basis for the illumination area of $10 \times 10\,\mu m^2$ and a numerical aperture of 0.6. The angle of illumination was set by writing a phase ramp of an appropriate slope and direction on the SLM. From a set of these measurements, we constructed transfer matrix $t(\eta; x, y)$ which is the complex-field amplitude of SPPs at point $\eta$ along the sampling line indicated in Fig. 3a for the illumination of a far-field wave at point $(x, y)$ located within the square area of the pattern[33,39]. (see the Supplementary Note 5 for the representative images and construction of the matrix.)

To confirm that the nanoholes far away from the sampling line can make a significant yet independent contribution to the control of the SPPs, we gradually increased the width of illumination $W$ and investigated the focusing of the SPPs on a target point. For the finite width of illumination $W \leq L$, a submatrix $t_1 = t(\eta; x, 0 \leq y \leq W)$ was chosen from the original matrix. A complex field map of illumination $\mathbf{E}_1$ was

identified that would maximize SPP intensity at a particular target point $\eta = \eta_1$ following the equation $\mathbf{E}_1 = t_1^{-1}\boldsymbol{\eta}_1$. Here, $\boldsymbol{\eta}_1$ is a vector whose element is unity at $\eta = \eta_1$ and zero otherwise. Setting an incident wave as $\mathbf{E}_1$ would lead to the constructive interference of SPPs originating from the nanoholes at the target point. At points $\eta \neq \eta_1$, SPPs would be added incoherently such that the net intensity would be much smaller than at the target point. The ratio of intensity between the focus point and the background will correspond to the effective channel number.

We experimentally demonstrated the focusing of SPPs by writing the phase map of $\mathbf{E}_1$ on the SLM and recording the complex field map of the SPPs at the camera. Experiments were performed by increasing $W$ at intervals of $1\,\mu m$. Representative images are shown in Fig. 3b–e for $W = 2, 5, 8$ and $10\,\mu m$, respectively. The boundary of the 2D pattern is indicated by a white dashed box, and the area of far-field illumination as a red rectangular box. The SPPs were clearly focused at the target point, and the focus became progressively more distinct as the width of illumination increased. From the line profiles along the sampling line (Fig. 4a), we observed that the intensity at the target point significantly increased. The intensity of the SPPs at the target point was plotted as a function of $W$ in Fig. 4b, from which we observed that the intensity increased by almost 100 times at the full width of illumination in comparison with $W = 1\,\mu m$. This confirms that the SPPs arising from nanoholes far away from the sampling line made a significant contribution to the control.

To prove that the effective channel number had increased, we assessed $N_{eff}^{2D}$ by measuring the signal-to-noise ratio of SPP focusing. As shown in Fig. 4c, the signal-to-noise ratio increased from $N_{eff}^{2D} = 2$ for $W = 1\,\mu m$ to $N_{eff}^{2D} = 40$ for $W = 10$ $\mu m$. Considering that $W = 1\,\mu m$ is asymptotically 1D slab, this corresponds to a $\alpha = N_{eff}^{2D}/N^{1D} = 20$ fold increase in the channel number, confirming the effectiveness of disordered nanoholes in enhancing information transfer capacity. On the other hand, the enhancement factor $\alpha$ was larger than the theoretical expectation. From the analysis of the measured transfer matrix, the characteristic length $l_c$ was measured to be $\sim 1\,\mu m$ (Fig. 4d). Since the width of the 2D disordered pattern used in the

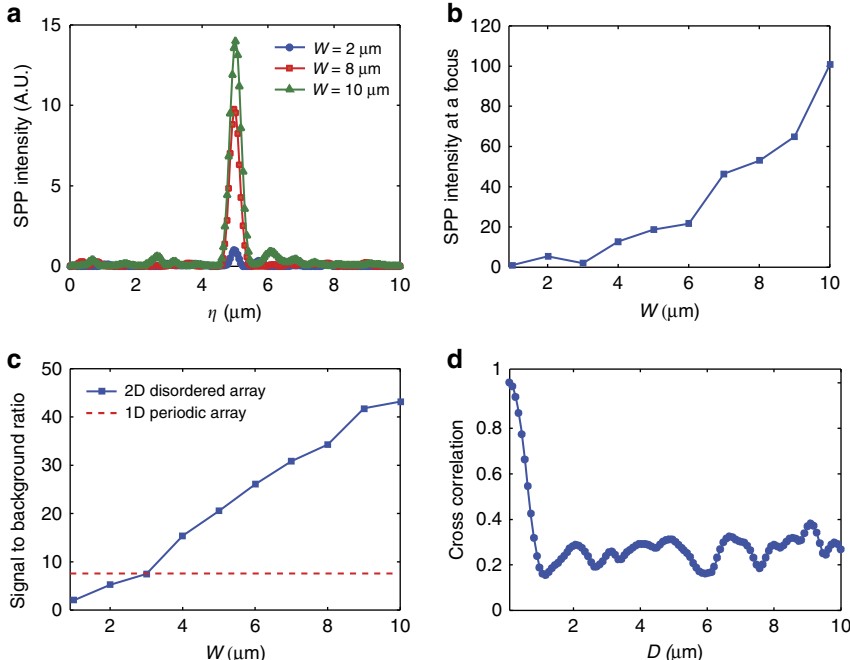

**Figure 4 | Demonstration of the increased channel number.** (**a**) Line profiles of the SPP intensity along the sampling line for different widths of illumination. (**b**) The intensity of SPPs measured at the target point as a function of the width of illumination. The intensity was normalized at $W = 1\,\mu m$. (**c**) Effective channel number $N_{\rm eff}^{\rm 2D}$ determined by the ratio of intensity between the target point and the background with respect to the width of illumination (square dots). The red dashed line indicates the signal-to-noise ratio for 1D periodic nanoholes. (**d**) Normalized cross-correlation of the SPPs originating from the segment of nanoholes at $0 \leq y \leq 100\,nm$ with those from $D \leq y \leq D + 100\,nm$ calculated from the measured transfer matrix (see Supplementary Note 2 for the detailed procedure).

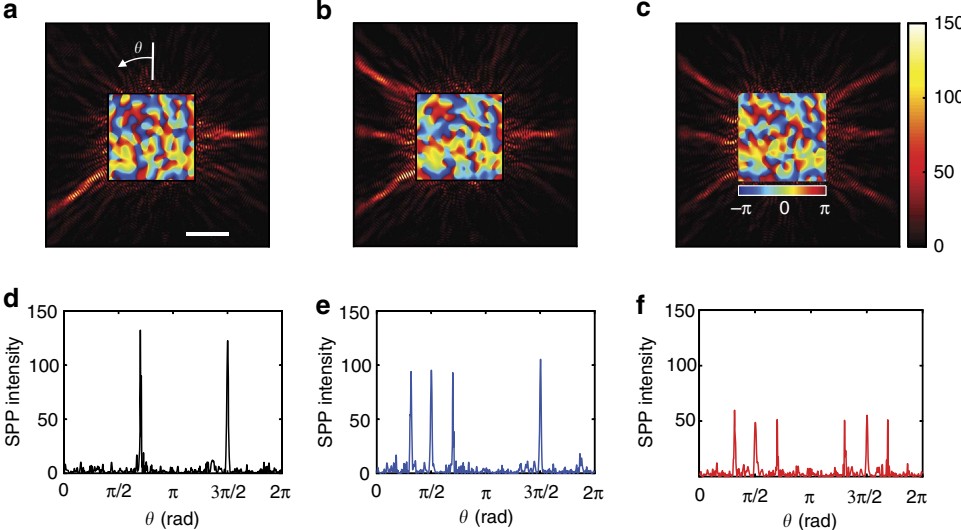

**Figure 5 | Experimental demonstration of a MIMO network using the focusing of SPPs at multiple spots.** (**a**–**c**) SPP maps after focusing at two, four and six spots, respectively. Vertical colour bar: intensity of the SPPs in an arbitrary unit. The square colour maps at the centre of the images indicate the phase maps of the far-field illumination incident to the pattern of nanoholes. Horizontal colour bar: phase in radians. Scale bar, 5 μm. (**d**–**f**) The intensity of SPPs along the sampling line with respect to the angle $\theta$ indicated in (**a**) for the cases of (**a**–**c**) respectively.

experiment was $L = 10\,\mu m$, the theoretically expected enhancement factor is $\alpha_{\rm th} \approx 9$ (Fig. 1e). The discrepancy is mainly due to the relatively imperfect control of incident waves at small $W$. The shaping of incident waves at a narrow width causes diffraction which interacts with neighbouring holes and generates unwanted SPPs. As a relevant control experiment, we prepared a 1D array of nanoholes and performed the same focusing

experiment (see the Supplementary Note 5). Because there were no neighbouring holes along the $y$-direction for the 1D pattern, the diffraction from the illumination does not generate SPP noise. The experimentally measured signal-to-noise ratio for the 1D pattern was $\sim 7.6$ (the dashed line in Fig. 4c). Therefore, the experimentally observed channel enhancement factor from 1D to 2D patterns was about $\alpha_{\rm exp} \approx 6$. The discrepancy between

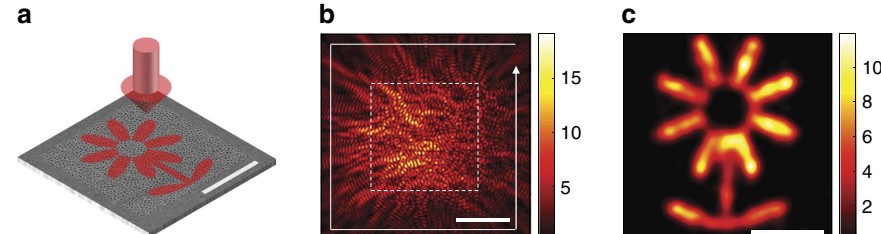

**Figure 6 | Experimental demonstration of the delivery of a 2D far-field image to a 1D SPP sampling line.** (**a**) A far-field illumination containing a flower pattern was projected to the patterned area on Au film. Scale bar, 5 μm. (**b**) Experimentally measured SPP map for the illumination of the flower pattern. The dashed white square indicates an area where the flower pattern was projected. SPPs were sampled along a white line forming a square of side length 17.3 μm. Colour bar: amplitude in an arbitrary unit. Scale bar, 5 μm. (**c**) Intensity map of the reconstructed image projected through the far-field illumination from the SPPs sampled along the white line in (**b**) Colour bar: intensity in an arbitrary unit. Scale bar, 2 μm.

$\alpha_{\mathrm{exp}}$ and $\alpha_{\mathrm{th}}$ arises mainly due to an approximate residual correlation of 20% (Fig. 4d). When this residual correlation was accounted for (see the Supplementary Note 2 for more details), the theoretically expected enhancement factor was ∼6 (the red dots in Fig. 1e) which was in excellent agreement with the experimental observation.

**Experimental demonstration of a MIMO network**. With the increased channel number, we can not only switch incoming far-field waves to individual output points at an improved signal-to-noise ratio, but also send them to multiple different spots at the same time. The signal-to-noise ratio of the individual spots in multiple-spot focusing is approximated by the signal-to-noise ratio of single-spot focusing divided by the number of target spots. In effect, the signal-to-noise ratio is shared by multiple spots. Therefore, the transmission channel number to $N_{\mathrm{eff}}^{\mathrm{2D}}$ is a critical factor to determine the number of output channels to which information can be transferred with enough signal-to-noise ratio. From the measured transfer matrix, we identified proper incident wave $\mathbf{E}_2$ using $\mathbf{E}_2 = t^{-1}\boldsymbol{\eta}_2$, where $\boldsymbol{\eta}_2$ is a unit-amplitude vector whose elements are zero except for the target spots. By writing the phase map of $\mathbf{E}_2$ on SLM, we experimentally demonstrated the focusing of SPPs at multiple different spots. The SPP maps are shown in Fig. 5a–c for two, four and six spots, respectively. The square maps displayed at the centre of the images indicate the phase maps of the far-field illumination written on the SLM. Clean focusing was observed even for six spots due to the enhanced effective channel number. To quantitatively assess the signal-to-noise ratio of the focusing, we plotted the intensity of the SPPs along the circle indicated with respect to $\theta$ in Fig. 5a. As shown in Fig. 5d–f, the intensity of the individual spots decreased when the number of target spots increased. Because the background noise remained approximately the same, signal-to-noise ratio progressively decreased. The measured signal-to-noise ratio was 20.8, 12.8 and 7.3 for the two-, four- and six-spot focusing, respectively.

**Experimental demonstration of image delivery**. Finally, we demonstrated the delivery of far-field 2D image information to 1D SPP output channels. As shown in Fig. 6a, a far-field wave $\mathbf{E}_t$ containing an amplitude pattern resembling a flower was projected onto the disordered array of nanoholes. The SPPs generated by this illumination were sampled along the solid white line forming a square (Fig. 6b). The measured SPPs formed vector $\eta_t$, from which we identified $\mathbf{E}_t$ using $\mathbf{E}_t = t^{-1}\boldsymbol{\eta}_t$. To reduce the noise of reconstruction, we averaged 400 far-field waves with the same amplitude pattern of the flower, but with different phase patterns. Figure 6c shows the reconstructed image

which faithfully reproduced the original flower pattern. This confirms that the decorrelation effect of the disordered array of nanoholes enabled the delivery of 2D image information projected to each and every point within the disordered pattern to the 1D SPP sampling line.

## Discussion

We have presented a 2D disordered array of nanoholes on a thin metal film as a MIMO plasmonic switching device. By exploiting the decorrelation of plasmonic waves due to their random multiple scattering, we could convert nanoholes encompassing a 2D area as independent antennas. We demonstrated the transfer of more than 40 far-field input channels to the SPPs which is about an order of magnitude increase in the number of transmission channels with respect to periodically ordered 2D devices. The use of lower loss material than gold such as silver is expected to increase channel capacity even further (see Supplementary Note 6). With the increased transmission channel number, we implemented the simultaneous control of 6 SPP channels at high signal-to-noise ratios. The additional experiments of delivering a 2D image embedded in far-field waves to the SPPs sampled along a 1D line confirmed that the nanoholes distributed across the 2D area indeed acted as independent transmission antennas.

Our method of exploiting the disorder to minimize the effect of the dimensional reduction from far-field waves to surface waves and therefore maximizing the deliverable far-field input channels to the plasmonic output channels will expedite the use of plasmonics in optoelectronic devices[5–11]. In particular, we expect that the successful integration of the proposed switching device with multiple electrical circuits will lead to a dramatic increase in the processing speed of complex computational tasks. For the real practices, additional steps may be necessary such as increasing the speed of channel control which can be done by replacing a spatial light modulator with the combination of a high-speed beam scanning device and multiple static phase masks. In the long run, high-end microprocessors can take the advantage of the proposed type of a high-throughput switching device and help a wide range of scientific activities relying on heavy computation.

**Data availability**. All relevant data are available from the authors.

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

## Acknowledgements

This research was supported by IBS-R023-D1 and the Global Frontier Project (2014M3A6B3063710) through the National Research Foundation of Korea (NRF) funded by the Ministry of Science, ICT and Future Planning. It was also supported by the Korea Health Technology R&D Project (HI14C0748) funded by the Ministry of Health and Welfare, Republic of Korea.

## Author contributions

Wonjun C. and Wonshik C. conceived the idea and designed the experiments. Y.J. and Wonjun C. carried out the measurements. Wonjun C., Y.J. and Wonshik C. analysed the data. Y.J., J.A., E.S. and Y.M.J. fabricated the samples. Q.-H.P. helped data interpretation. Wonjun C., Y.J. and Wonshik C. prepared the manuscript, and all authors contributed to finalizing the manuscript.

## Additional information

**Competing financial interests:** The authors declare no competing financial interests.

