## [Peer Review File · Nature Communications]

Reviewers' comments:

Reviewer #1 (Remarks to the Author):

The manuscript entitled 'Control of randomly scattered surface plasmon polaritons for multiple-input and multiple-output plasmonic switching devices' is an interesting work. It is important to realize impact on-chip plasmonic devices which can convert far-field waves to surface plasmon and delivery the input channels to the plasmonic output channels. And in this work, the authors demonstrated the transfer of more than 40 far field input -channels to the SPPs. I think the basic physics of this work is clear. The overall experimental and simulation results are very nice. I thus recommend it for publication in Nature Communications with some r comments:

1. Is the fill factor linked to the performance of the proposed device? Is there a reason why there should be 12 %, not 5%, 15% or other values? The authors should analyze the effect of fill factor.
2. The authors should add some details on how to calculate the cross correlation in Fig. 4d.
3. There is big discrepancy between the theoretic model and the experimental result. Is it possible to come up with a more accurate model? Additionally, to convince the readers, extra experiment with different L (such as 5 μm and 10 μm) should be conducted.
4. Please describe the design method for MIMO network with different spots. Is it a general method? What if I need 8 spots in a sampling line along x axis?
6. Line99, 'we experimentally focused the SPPs at target spots (Fig. 4c)'. However, Fig. 4c discusses the SNR for different W. It seems it is not correct (also see Line127 and Line139). Please double check it.

Reviewer #2 (Remarks to the Author):

The paper by Wonjun Choi, Yonghyeon Jo et al. is based on a very interesting idea, the use of disorder at it's best: increasing the number of possibilities. Unfortunately the use of disorder is experimentally and theoretically challenging. Good experimental proofs of concept are rare. This is why I think this manuscript, which succeed at showing an increase of the accessible bandwidth of surface plasmon polariton (SPP), is worth publishing in a journal with a wide audience. But I have few comments and questions that I would like the authors to clarify.

- Is there a reason to use a rectangular slab spanning the entire length of the pattern (along x) ? I am not sure it is the best possible shape here, as the absorption length plays obviously an important role. Why was the 'infinite' slab geometry chosen, given the fact that transmission matrix and SLM work with almost any shape ?

- It seems that for small $m=L/l_c$, the 'theoretical' enhancement factor is almost equal to m. If the relation giving $T(L)$ is correct, and juggling from the parameter for l_a and l_t , it seems that the absorption dominates in the expression of $T(L)$. This simplification would also simplify the sum in equation (1).

- On absorption again, the wavelength chosen is obviously not the best one for gold, and gold is not the best plasmonic material in the visible. I understand easily why silver slabs were not used in the experiments. I understand a bit less why a longer wavelength was not used (is it due to the bandwidth of the SLM?). But why is there no simulation in the manuscript with better material and wavelength? This could demonstrate how far the technique can go.

- How could the authors confirm with certainty that there is no influence of the 'far-field' scattering of the SPP on their measurement (far-field waves that appear in the Fourier plane with a radius larger than k_0)? In the same frame of mind, the surface of the sample does not look flat at all, even out of the disordered array of nanoholes (fig 3a). Did the author make a control experiment

on a flat Au surface? The presence of scattering by the surface roughness, if it exists, does not alter the quality of the experiments, but might complicate the analysis and the comparison with the simulation.

- The paper is in general clearly written but some parts might be improved, like the discussion on the enhancement factor or part 1.2 of the supplementary section. I am not sure to understand what the authors call a "theoretical" analysis. I found no presentation of a theory in the manuscript, rather a model and some simulations. Therefore the discussion on the 'theoretical enhancement factor' is sometimes unclear to me.

We thank the reviewers for their constructive comments, which helped us to improve the quality of our manuscript. In this letter, we have addressed all the points raised by the reviewers. In particular, we performed additional experiments and numerical simulations for samples with various fill factors, sizes and materials to further support our theoretical model. A number of changes were made in the revised manuscript to accommodate the reviewers' points, and they were all highlighted in red. The additional data and analysis were added to the Supplementary Information.

Reviewer # 1

The manuscript entitled 'Control of randomly scattered surface plasmon polaritons for multiple-input and multiple-output plasmonic switching devices' is an interesting work. It is important to realize impact on-chip plasmonic devices which can convert far-field waves to surface plasmon and delivery the input channels to the plasmonic output channels. And in this work, the authors demonstrated the transfer of more than 40 far field input -channels to the SPPs. I think the basic physics of this work is clear. The overall experimental and simulation results are very nice. I thus recommend it for publication in Nature Communications with some comments:

We appreciate the reviewer for acknowledging the importance of our work and also providing us with highly constructive comments that helped us to improve the quality of the manuscript.

1. Is the fill factor linked to the performance of the proposed device? Is there a reason why there should be 12 %, not 5%, 15% or other values? The authors should analyze the effect of fill factor.

This is an important question that we should have discussed in the original manuscript. In fact, we had tested samples with different fill factors and observed that the optimal fill factor was around 12 %. We performed additional experiments to systematically investigate the effect of the fill factor. Figure R1 shows the enhancement factor of samples whose fill factors were ranging from 3 to 15 %. We observed that the enhancement factor was increased up to 12 %, and then decreased at 15 %.

Figure R1. Signal to background ratio, or the enhancement factor, for various fill factors. The size of the devices was $L = 10 \mu\text{m}$. For each device, enhancement factors were measured at different target spots along the sampling line to obtain their statistical distributions.

According to our theoretical model, the effective channel number can be described by the following relation:

$$N_{eff}^{2D} = \alpha N^{1D} = \frac{|\sum_{j=0}^{m-1} \sqrt{T(jl_c)}|^2}{\sum_{j=0}^{m-1} T(jl_c)} N^{1D}.$$

Here the fill factor affects to the enhancement factor N^{1D} per segment of $L \times l_c$ and the transmittance of SPPs $T(y)$ from holes located at y to the sampling line. In order to reach theoretical N^{1D} , the fill factor should be larger than approximately 3 % at the very least for the given experimental condition, and in fact the higher the better. Otherwise, there are too a small number of nanoholes that can convert far-field channels to those of SPPs. On the other hand, the increase in the fill factor results in the steep decrease of $T(y)$ due to the scattering loss, thereby reducing N_{eff}^{2D} .

But there exists additional important effect that is determined by the fill factor. The disordered array of nanoholes makes the contribution of SPPs from different segments independent. This was made clear in the cross-correlation plot shown in Fig. 4d. Even with the use of the disordered array, however, there exists residual correlation which reduces the effective number of converted channels. At higher fill factor, the residual cross-correlation among segments is smaller. As a result, the SPPs from different segments are more independent such that the enhancement factor approaches to the theoretical expectation. Accounting for all these three effects that the fill factor gives rise to, there exists a certain fill factor for the optimum channel conversion efficiency, which was about 9-12 % for our experimental condition.

We added the following sentence to the revised manuscript and also to the supplementary section 5.

“Individual holes measuring 100 nm in diameter filled an $L \times L = 10 \times 10 \mu\text{m}^2$ area with a fill factor ranging from 3 to 15 %. Since the channel conversion efficiency was the best at 12% fill factor in our experiment (See Supplementary Section 5), all the data shown in the main text used the 12 % samples.”

2. The authors should add some details on how to calculate the cross correlation in Fig. 4d.

Due to the limitation of space in the figure caption, only a brief explanation was given for Fig. 4d. The data points in the figure were the magnitude of the cross-correlation of the SPPs originating

from the segment of nanoholes at $0 \leq y \leq 100$ nm with those from $D \leq y \leq D + 100$ nm. The mathematical procedure is given in detail in the following. The SPP field at the sampling line $\boldsymbol{\eta}_1(\eta)$ can be calculated for the illumination of far field wave \mathbf{E}_n at the slab $0 \leq y \leq 100$ nm from the measured transmission matrix t :

$$\boldsymbol{\eta}_1(\eta) = t(\eta; x, 0 \leq y \leq 100 \text{ nm}) \times \mathbf{E}_n(x, 0 \leq y \leq 100 \text{ nm}).$$

Here \mathbf{E}_n can be arbitrary, but we chose it to be normally incident plane wave such that the amplitude is the same for all the elements within the segment defined by $(0 \leq x \leq L, 0 \leq y \leq 100 \text{ nm})$. Likewise, the SPP field $\boldsymbol{\eta}_2(\eta)$ generated by the far field illumination \mathbf{E}_n at the slab $D \leq y \leq D + 100$ nm can be calculated by the following calculation:

$$\boldsymbol{\eta}_2(\eta) = t(\eta; x, D \leq y \leq D + 100 \text{ nm}) \times \mathbf{E}_n(x, D \leq y \leq D + 100 \text{ nm}).$$

Then, the normalized cross-correlation of the two SPP fields was calculated by the following equation:

$$C(D) = \frac{\langle \boldsymbol{\eta}_1^* \cdot \boldsymbol{\eta}_2 \rangle}{\|\boldsymbol{\eta}_1\| \cdot \|\boldsymbol{\eta}_2\|}.$$

We added this detailed description for the Fig. 4d to the Supplementary Section 6 and added this following sentence to the figure caption.

“(see Supplementary Section 6 for the detailed procedure)”

3. There is big discrepancy between the theoretic model and the experimental result. Is it possible to come up with a more accurate model? Additionally, to convince the readers, extra experiment with different L (such as 5 μm and 10 μm) should be conducted.

The enhancement factor of channel number that our theoretical model,

$$\alpha = \frac{N_{eff}^{2D}}{N^{1D}} = \frac{|\sum_{j=0}^{m-1} \sqrt{T(jl_c)}|^2}{\sum_{j=0}^{m-1} T(jl_c)},$$

predicts for the given experimental parameters ($L = 10 \mu\text{m}$, $l_c = 1 \mu\text{m}$, $l_a \approx 2.4 \mu\text{m}$, and $l_t \approx 7.2 \mu\text{m}$) was $\alpha_{th} \approx 9$. On the other hand, the experimentally measured enhancement factor presented in Fig. 4c was $\alpha_{exp} \approx 6$. This discrepancy was mainly due to the residual correlation of SPPs originating from different segments in the disordered array of nanoholes. As shown in Fig. 4d, this residual correlation was measured to be about 0.2. We accounted for this correlation in estimating α , and the modified model predicts the enhancement factor of around 6 (red dots in Fig. 1e), a good agreement with the experimental results.

In the original manuscript, it was only briefly mentioned for the way to incorporate the residual correlation into the modified model. We now explained the detailed procedure to include its effect on α .

We numerically generated a transfer matrix $t_{num}(\eta; x, y)$ using random matrix theory in which the columns of the matrix are completely orthogonal with respect to each other. We then applied the attenuation of the SPPs depending on y following the transmittance function $T(y)$. The same vector was added to each column in such a way that there exist 20 % residual cross-correlation among columns of different y . Finally, we used this matrix to calculate the enhancement factor of enhancement, which is shown as red dots in Fig. 1e.

We also added this discussion to the Supplementary Section 7 in the revised manuscript, and added the following sentence to the caption of Fig. 1.

“(See Supplementary Section 7)”

Following the reviewer’s suggestion, we also performed additional experiments for $L = 5 \mu\text{m}$ sample at the same 12 % fill factor as $L = 10 \mu\text{m}$ sample (Fig. R2). The slope was smaller at $L = 5 \mu\text{m}$ than $L = 10 \mu\text{m}$ because the enhancement factor N^{1D} per segment of $L \times l_c$ is smaller. In addition, there were only 5 effective segments (the effective number of segment is given by $m = \lfloor L/l_c \rfloor$) in $L = 5 \mu\text{m}$ sample while there were 10 in $L = 10 \mu\text{m}$ sample. Due to these combined effects, the enhancement factor for $L = 5 \mu\text{m}$ sample was smaller than that of $L = 10 \mu\text{m}$. But the behavior of the increase of channel conversion efficiency was similar.

Figure R2. Enhancement factor for $L = 5 \mu\text{m}$ and $L = 10 \mu\text{m}$ samples as a function of the width of illumination. Both samples have the same 12 % fill factor.

We added the results of these additional experiments to the Supplementary Section 8.

4. Please describe the design method for the MIMO network with different spots. Is it a general method? What if I need 8 spots in a sampling line along x axis?

The general layout of MIMO network used in wireless communications is described below.

Figure R3. Typical layout of multiple-input multiple-output network. T_1, T_2, \dots are the transmitters and R_1, R_2, \dots are the receivers. The h_{ij} stands for the amplitude transmittance from i^{th} transmitter to the j^{th} receiver.

Typically, a number of transmitters (T_1, T_2, \dots, T_n) and receivers (R_1, R_2, \dots, R_m) are spatially multiplexed to enhance channel capacity. Spatial multiplexing is made possible due to the multiple scattering of waves emanating from transmitters on their way to receivers by means of random reflections from walls and clouds. In our case, the disordered array of nanoholes plays a similar role. In order to decode the multiplexed signals, the inversion of the transfer matrix H , whose elements h_{ij} are the amplitude transmittance from i^{th} transmitter to j^{th} receiver, is used. In the real-world wireless communications, null data is sent from transmitters to receivers to measure H prior to the transfer of information. This is exactly the same as our measurements of transfer matrix.

For a comparison, we suggest a possible optoelectronic network in Fig. R4 that our device of the disordered array of nanoholes can be integrated with. The disordered array of nanoholes may sit in the middle, and multiple electronic devices (output devices 0-6 in this example) can be located around. The multiple input device, which is the spatial light modulator in our study, controls far-field channels and therefore corresponds to the transmitters in MIMO network. Likewise, the electronic devices located around the disordered array of nanoholes are equivalent to the receivers. Therefore, the number of spots to be used for the control will depend on the number of electronic devices to be connected to. And the maximum number of devices that can be connected to will be determined by the channel capacity, which is 40 in our case.

Figure R4. The exemplary layout of the optoelectronic MIMO network by using the disordered array of nanoholes as a switching device.

In many aspects, our device has similar layout with the conventional MIMO network, but there exists important difference in the types of signals. Our device mediates far-field waves propagating in 3D to SPPs propagating in 2D, while the ordinary MIMO network concerns far-field waves for both transmitters and receivers.

In order to enlighten the readers for the way our device can form MIMO networks in optoelectronics, we added this discussion to the Supplementary Section 9. We also added the following sentence to the revised manuscript.

“(see Supplementary Section 9 for the possible layout of the optoelectronic networks)”

5. Line99, ‘we experimentally focused the SPPs at target spots (Fig. 4c)’. However, Fig. 4c discusses the SNR for different W . It seems it is not correct (also see Line127 and Line139). Please double check it.

We appreciate the reviewer for the careful reading of our manuscript. In fact, the sentence in the line 99 refers to Fig. 3, not Fig. 4c. The referencing of line 127 to Fig. 4d for l_c was correct. And the line 139 referring Fig. 4a to the T(L) is rather indirect. We therefore removed the word ‘Fig. 4a’ and left only the referencing to the supplementary information. All these changes were made in the revised manuscript.

Reviewer # 2

1. The paper by Wonjun Choi, Yonghyeon Jo et al. is based on a very interesting idea, the use of disorder at its best: increasing the number of possibilities. Unfortunately, the use of disorder is experimentally and theoretically challenging. Good experimental proofs of concept are rare. This is why I think this manuscript, which succeed at showing an increase of the accessible bandwidth of surface plasmon polariton (SPP), is worth publishing in a journal with a wide audience. But I have few comments and questions that I would like the authors to clarify.

We appreciate the reviewer for acknowledging the importance of our work. The reviewer’s comments and suggestions were highly valuable. We addressed each and every point, which has resulted in improving the integrity of our study.

2. Is there a reason to use a rectangular slab spanning the entire length of the pattern (along x)? I am not sure it is the best possible shape here, as the absorption length plays obviously an important role. Why was the ‘infinite’ slab geometry chosen, given the fact that transmission matrix and SLM work with almost any shape?

In order to answer this question, we need to discuss the feasible optoelectronic MIMO network that our device can be a part of. As described in detail to the comment #4 of the Reviewer #1, we envision that an optoelectronic MIMO network can be formed in the form in Fig. R5.

Figure R5. Potential layout of the optoelectronic networks using the disordered array of nanoholes as a switching device. The switching device can be located either in the middle (a) or outer rim (b) of the networks.

The disordered array of nanoholes may be located at the center to convert incoming far-field waves to the SPPs and switch optical information to the multiple electrical output devices. The shape of the disordered array doesn't need to be square, but it may be better for the nanoholes to fill a closed shape of a two-dimensional figure. This will ensure the effective connection to as many electronic devices as possible. For the simplicity of analysis, we considered rectangular shape of finite size in both x and y directions. As the reviewer mentioned, SPPs originating from the middle of the array far from the sampling line contribute less for the channel conversion efficiency than those from the proximity of the sampling line. But our study proved that nanoholes in the middle can still make a significant contribution.

In fact, we also considered a disordered array of nanoholes forming an arena in the middle. In this case, the electronic devices can be located in this central arena. Then, the number of nanoholes contributing to the channel conversion can be larger than before as more holes are near the output devices. However, the space for electronic devices is confined only to the arena, which limits the number of electronics devices to be connected.

3. It seems that for small $m=L/l_c$, the 'theoretical' enhancement factor is almost equal to m . If the relation giving $T(L)$ is correct, and judging from the parameter for l_a and l_t , it seems that the absorption dominates in the expression of $T(L)$. This simplification would also simplify the sum in equation (1).

As the reviewer pointed out, the enhancement factor is almost equal to m for small m because the attenuation of $T(y)$ is relatively weak for the SPPs generated at the segments near the sampling line. However, the attenuation in $T(y)$ is not negligible for large m . Therefore, the enhancement factor becomes smaller than m as the size of the device is increased.

The attenuation of $T(y)$ is determined by l_a and l_t by the relation, $T(y) = (l_t/(y + c))\exp(-y/l_a)$. Now the question is to what degree the l_t contributes to the enhancement factor. If we consider only l_a , the transmittance is given simply by $T_1(y) = \exp(-y/l_a)$. As shown in Fig. R6a, $T(y)$ is smaller and decays faster than $T_1(y)$. As a consequence, the enhancement factor is smaller when l_t is included (Fig. R6b). However, as the reviewer conjectured, the

simple case of $T_1(y)$ agrees quite well up to $m = 4$ with the case of $T(y)$. Therefore, the simple inclusion of l_a should be good enough for small m . Indeed, the effect of l_t is pronounced mainly at large m .

Figure R6. The effects of absorption and scattering losses to the channel enhancement factor. **a**, Transmittance of SPPs for a distance y . Blue dots: FDTD simulation, blue curve: $T(y)$, red curve: $T_1(y)$. **b**, Theoretical estimation of the channel enhancement factor as a function of the size of the device for the cases of $T(y)$ (blue dots) and $T_1(y)$ (red dots).

We added the discussion above to the Supplementary Section 10, and inserted the following sentence into the revised manuscript.

“In general, l_a plays a major role at small y , and l_t becomes more important at large y (see Supplementary Section 10)”.

4. On absorption again, the wavelength chosen is obviously not the best one for gold, and gold is not the best plasmonic material in the visible. I understand easily why silver slabs were not used in the experiments. I understand a bit less why a longer wavelength was not used (is it due to the bandwidth of the SLM?). But why is there no simulation in the manuscript with better material and wavelength? This could demonstrate how far the technique can go.

We greatly appreciate the reviewer’s suggestion. The loss of SPPs is one of the governing factors for the channel enhancement factor. Therefore, the use of silver layer as a host medium and illumination of light wave with longer wavelength will be beneficial to the channel conversion efficiency. Following the reviewer’s suggestion, we performed additional numerical simulation for a sample made of a silver layer using the wavelength of light source $\lambda = 800$ nm. For a direct comparison with the gold sample, the fill factor of the nanoholes was set to 12 %. As shown in Fig. R7, however, the enhancement factor of the silver sample (black dots) was smaller than that of the gold sample (blue dots). It turned out that the use of longer wavelength did more harm than good. The correlation length l_c measured by the width of the cross-correlation curve was made longer because the effective diffraction limit spot along the y -direction was larger. Therefore, the effective number of slabs $m = [L/l_c]$ was smaller for the silver sample at $\lambda = 800$ nm. Another disadvantage of using longer wavelength is that the channel number N^{1D}

per segment of $L \times l_c$ was smaller as the effective number of diffraction limit spots was reduced along the x -direction in the slab. This can be seen at $W = 0.5 \mu\text{m}$ in Fig. R7 in which the signal to noise ratio for Au at $\lambda = 620 \text{ nm}$ was larger than that for Ag at $\lambda = 800 \text{ nm}$.

We therefore performed additional simulation for the same silver sample, but at the same wavelength $\lambda = 620 \text{ nm}$ as gold sample. As shown by the red dots in Fig. R7, the enhancement factor of the silver sample was now larger than that of the gold sample. When W was increased, the increase in the enhancement factor was rather linear in the silver sample while the slope of enhancement factor was decreased in gold sample. This was mainly due to the reduced loss of SPPs in the silver sample. These additional studies suggest that the use of low loss sample is preferable for the maximal channel transfer.

Figure R7. Numerical study comparing silver layer with gold layer. **a**, Signal to noise ratio with respect to W predicted from the calculated $T(y)$. Black and red dots are from silver samples at the wavelength of 800 nm and 620 nm, respectively. Blue dots are from the gold sample at the wavelength of 620 nm. **b**, Signal to noise ratio measured by the numerical focusing. Black, red and blue dots indicate the same conditions as those specified in **a**. The discrepancy between the measurements and predictions was due to the residual correlation discussed in Supplementary Section 1.3.

We added this discussion on the comparison between silver and gold layers to the Supplementary Section 11. Also, the following sentence was added to the conclusion section of the revised manuscript to summarize this further investigation.

“The use of lower loss material than gold such as silver is expected to increase channel capacity even further (see Supplementary Section 11).”

5. How could the authors confirm with certainty that there is no influence of the ‘far-field’ scattering of the SPP on their measurement (far-field waves that appear in the Fourier plane with a radius larger than k_0)? In the same frame of mind, the surface of the sample does not look flat at all, even out of the disordered array of nanoholes (fig 3a). Did the author make a control experiment on a flat Au surface? The presence of scattering by the surface roughness, if it exists, does not alter the quality of the experiments, but might complicate the analysis and the comparison with the simulation.

The reviewer raised two important points. Regarding the first point on the influence of the residual far-field waves, we performed a quantitative analysis on the relative intensities of the far-

field waves and SPPs. The typical intensity map taken at the back aperture of the objective lens (the inset image in Fig. 2) shows that there were far-field waves as well as SPPs in the leakage radiation microscope. As explained in the main text, we placed a beam block to minimize the intensity of the far-field waves at the camera. Figure R8a shows the typical field map recorded at the camera. The angular intensity distribution of this detected wave is shown in Fig. R8b in which the bright ring that corresponds to k_{SPP} was the dominant signal. From this data, we analyzed the relative intensity between SPPs and far-field waves, and observed that the total residual far-field waves were about 10 % of SPPs. This suggests that the far-field waves contributed little to the control of SPPs.

Figure R8. Comparison between SPPs and far-field waves. **a**, Typical intensity map measured by the interferometric leakage radiation microscope. Color bar, intensity in arbitrary unit. Scale bar, 5 μm . **b**, Fourier transform of the complex field map in a, which corresponds to the angular distribution of detected field. θ indicates azimuthal angle of scattered wave. Bright ring corresponds to k_{SPP} . Color bars, intensity in arbitrary unit.

Regarding the second point on the contribution of the surface roughness, we performed additional experiments for the samples with no nanoholes. Although there existed SPPs from the surface, its intensity was much smaller than those from nanoholes. Figure R9 shows the optimized intensity at the target as a function of the width of illumination for the sample with no nanoholes (square dots) in comparison with the sample of 12 % fill factor (circular dots). According to these measurements, the contribution of the surface roughness was measured to be less than 1 % at the full width of illumination. Therefore, we could confirm that SPPs from the nanoholes were the main source of signals in our experiment.

Figure R9. Control experiment using the sample with no nanoholes. The intensity of the optimized spot was measured as a function of the width of illumination (square dots). As a comparison, the same data is shown for the sample with the 12 % fill factor of nanoholes (circular dots). The two curves were displayed at the same scale for the direct comparison of their intensities.

We added the following sentence to the revised manuscript to justify the validity of our experiments. The additional analysis and experimental results were covered in Supplementary section 12.

“According to our analysis, the total intensity of the residual far-field waves was about 10 % of the total SPP intensity (Supplementary Section 12). The surface roughness of the metal layer also generated unwanted SPPs and far-field waves, but their contribution was measured to be less than 1 % of the total detected wave.”

6. The paper is in general clearly written but some parts might be improved, like the discussion on the enhancement factor or part 1.2 of the supplementary section. I am not sure to understand what the authors call a “theoretical” analysis. I found no presentation of a theory in the manuscript, rather a model and some simulations. Therefore the discussion on the ‘theoretical enhancement factor’ is sometimes unclear to me.

We agree with the reviewer in that we presented a theoretical model that explains the conversion of far-field channels to those of SPPs rather than theoretical analysis. We therefore unified the terminologies as theoretical model.

REVIEWERS' COMMENTS:

Reviewer #1 (Remarks to the Author):

In light of the authors changes to the manuscript, I believe that the work has improved and will be suitable for publication.

Reviewer #2 (Remarks to the Author):

The authors have carefully replied to all the comments I had on the previous version of the manuscript. Both manuscript and supplementary material section have been amended and corrected in order to improve the overall quality and clarity of the paper. Therefore I think the present version of the manuscript is suitable for publication in Nature Communications.